# Security and Cryptographic Challenges for Authentication Based on Biometrics Data

**Stefania Loredana Nita [1], Marius Iulian Mihailescu [2,\*] and Valentin Corneliu Pau [3]**

[1] Department of Computer Science, University of Bucharest, 030018 Bucharest, Romania; stefanialoredananani@gmail.com

[2] Department of Data Systems, Royal Caribbean, Miami, FL 33132, USA

[3] The Academy of Romanian Scientists, 050094 Bucharest, Romania; v_pau@utm.ro

\* Correspondence: mihmariusiulian@gmail.com; Tel.: +40-740030310

**Abstract:** Authentication systems based on biometrics characteristics and data represents one of the most important trend in the evolution of the society, e.g., Smart City, Internet-of-Things (IoT), Cloud Computing, Big Data. In the near future, biometrics systems will be everywhere in the society, such as government, education, smart cities, banks etc. Due to its uniqueness, characteristic, biometrics systems will become more and more vulnerable, privacy being one of the most important challenges. The classic cryptographic primitives are not sufficient to assure a strong level of secureness for privacy. The current paper has several objectives. The main objective consists in creating a framework based on cryptographic modules which can be applied in systems with biometric authentication methods. The technologies used in creating the framework are: C#, Java, C++, Python, and Haskell. The wide range of technologies for developing the algorithms give the readers the possibility and not only, to choose the proper modules for their own research or business direction. The cryptographic modules contain algorithms based on machine learning and modern cryptographic algorithms: AES (Advanced Encryption System), SHA-256, RC4, RC5, RC6, MARS, BLOWFISH, TWOFISH, THREEFISH, RSA (Rivest-Shamir-Adleman), Elliptic Curve, and Diffie Hellman. As methods for implementing with success the cryptographic modules, we will propose a methodology which can be used as a how-to guide. The article will focus only on the first category, machine learning, and data clustering, algorithms with applicability in the cloud computing environment. For tests we have used a virtual machine (Virtual Box) with Apache Hadoop and a Biometric Analysis Tool. The weakness of the algorithms and methods implemented within the framework will be evaluated and presented in order for the reader to acknowledge the latest status of the security analysis and the vulnerabilities founded in the mentioned algorithms. Another important result of the authors consists in creating a scheme for biometric enrollment (in Results). The purpose of the scheme is to give a big overview on how to use it, step by step, in real life, and how to use the algorithms. In the end, as a conclusion, the current work paper gives a comprehensive background on the most important and challenging aspects on how to design and implement an authentication system based on biometrics characteristics.

**Keywords:** machine learning; chaos-based cryptography; Hadoop; data clustering; biometrics

---

## 1. Introduction

Biometric represents a science through which a system can identify in a unique way the individual based on his physiological (face, fingerprint, iris, hand geometry, retina, etc.) and behavioral (voice, gait, signature, keystroke, etc.) traits. Due to the rich of applications and systems that occur at this moment on the market, the authentication technology is very widespread from e-commerce applications, door access, and smart city technologies to Internet-of-Things (IoT) technologies and

applications. Once these solutions where declared as providing better security in authentication, automatically issues with privacy preserving and data integrity has been found did not stop to occur. Indeed, biometrics systems become more convenient to be used as compared with different classic authentication systems such as token based (e.g., ID cars) or knowledge based (e.g., passwords) [1]. A biometric-based authentication system is using two different operating modes: identification and verification modes. These two different modes are very important as they represent security gaps, which are exploited by unwanted users (e.g., hackers/crackers).

Developing and implementing authentications modules for software and web applications, using biometric characteristics, can be a challenging task for normal (desktop) environments. If we are moving to cloud computing, map reduce, big data, and Hadoop technologies, we can experience real tasks that are difficult to accomplish with standard cryptographic primitive if they are not improved and certain extra security parameters added. As we saw, we can improve the authentication process by using a multi-factor authentication scheme as we already proposed in [1]. Further, if we want to step out and to move to modern environments, such as Hadoop with Data Clustering (Section 3), we need to focus on the security aspects of these environments [2–5].

The general goal of the paper is to provide a security framework for developing secure software and applications in which the authentication process based on biometrics is done in a secure way. Providing high-level security mechanism is the most interesting and challenging part. The aim is to make tests and practical implementations in order to check if the biometric characteristics can be used in a complex encryption/decryption process for cloud computing and Hadoop environment.

In order to accomplish the goal, we focus on four other major tasks (sub-goals):

- Creating a framework containing cryptographic modules. The cryptographic modules are implemented in C#, Java, C++, Python, and Haskell;
- Implementing two cryptographic modules, (1) modern cryptographic module and (2) machine learning approach cryptography module. The second module contains algorithms with auxiliary comparisons over unencrypted and encrypted inputs, creating statistical analysis of how strong the security of the inputs is;
- Creating a virtual laboratory for testing the algorithms and the framework. The requirements of the virtual laboratory are: Virtual Box virtual machine with Apache Hadoop, Biometric Analysis Tool, and 36 Virtual Box virtual machines which simulates the users trying to authenticate to the system and which represents the data cluster;
- Proposing a methodology which can be used as a how-to guide for how to implement the framework (Section 4).

The published results demonstrate that authentication of a person and the process of recognition of that person based on its biometric characteristic is a difficult problem to solve and to bring a solution to resolve the situation. According to our best knowledge in the field, there is no method published so far which can demonstrate the performance to guarantee the successful implementation in industrial and business solutions. The reason is based on three main obstacles: (1) definition of biometric characteristics accordingly; (2) the selection of features to be analyzed and encrypted properly; and (3) the processing methods which are not eligible to be fully trusted.

A successful solution to the mentioned problem of assuring the security of the authentication process in a desktop environment or cloud computing is very important. This is due to the fact that it can anticipate that the biometric characteristic will be used independent without any other security parameter (e.g., PIN code, password, or something that only the users knows or have as a physical device—token generator device).

The paper structure is as follows:

- Section 1—Introduction. This section will give a brief introduction about the current work by highlighting the main objectives, methods and obtained results. The chapter will give a short

background on how the main three phases—enrollment, verification and identification—are working in a software application which uses authentication based on biometrics.

- Section 2—Algorithms and Methods Used. The section present shortly which are the main algorithms and methods used for evaluation of security authentication process in a cloud and data clustered environment. The following topics are covered briefly: machine learning classification over biometric encrypted data; modern cryptography; techniques such as Naïve Bayes for classification of security incidents and rate of failing to authenticate with success, Hyperplane decision; data clustering aspects for Map Reduce covering the fixed-width clustering algorithm and FWC algorithm.
- Section 3—Biometrics and Authentication Mechanisms: How They Are Working. The section will go through the three main phases of a biometric system (enrollment, verification, and identification) with the goal of understanding how the authentication process is working and which are vulnerable points.
- Section 4—Results. In this section we will discuss about the current state of other similar works proposed by other authors by showing the main advantages and disadvantages.
- Section 5—Comparison with Other Proposed Methods and Discussion. The section will present in details the proposed solution and methodology showing the advantages and disadvantages compared with other methods. The section will demonstrate also how to design and implement software solution for desktop applications and for cloud computing with Apache Hadoop.
- Section 6—Conclusions. The section will cover the current objectives accomplished within the current paper.

## 2. Algorithms and Methods Used

In this section, we present two machine-learning techniques applied on biometric encrypted data: hyperplane decision and Naïve Bayes.

### 2.1. Preliminaries

2.1.1. Machine Learning Classification over Biometric Encrypted Data

"Machine learning is a field of computer science that gives the ability to learn without being explicitly programmed"—Samuel Arthur [6].

The below techniques have been applied and implemented in a software solution in order to simulate the methods on biometric data. The learning techniques implemented are described below. The solution software was implemented in. NET Framework 4.5 using C# and Microsoft SQL Server 2016. Due to the status of the software application, we cannot provide the source code in this work paper. For those who are interested about the application they can visit the web page https://www.researchgate.net/project/Biometrics-Analysis-Tool.

There are four types of learning techniques implemented are:

- Supervised learning is a type of inductive learning based on training sets, in which, the agent receives a set of inputs and their corresponding outputs. The task of the agent is to learn the links between every input and its corresponding output and to generate a template function that will be able to solve problems for new inputs.
- Semi-supervised learning. In this type of learning, the agent receives an incomplete training set.
- Unsupervised learning is not using training sets, but the agent needs to discover on its own different patterns in dataset.
- Reinforcement learning is a type of learning in which the training data is given as feedback for the agent, such that if its output is "good" it receives a reward, otherwise it receives a punishment. The target of the agent is to maximize its reward, providing better and better

outputs. The meaning of "good" output is different depending on the environment in which the agent is used [7].

- Classification represents a machine learning technique (included in supervised learning) in which the inputs are divided into two or more classes. The input is a *feature vector* $v = (v_1, \ldots, v_n) \in \mathbb{R}^n$ that will be classified by applying a classification function $f_m : \mathbb{R}^n \to \{x_1, \ldots, x_c\}$ on $v$, and the output is $x_{c^*} = f_m(v)$, where $c^* \in \{1, \ldots, c\}$; $x_{c^*}$ is the class in which $v$ falls, based on model $m$.

In this case, the feature vector will represent all the biometric data over which the classification data is applied.

$$v = (v_1, \ldots, v_n) \in \mathbb{R}^n$$
$$biometric\ vector = (b_v) = (b_{v_1}, \ldots, b_{v_n}) \tag{1}$$

$f_m : \mathbb{R}^n \to \{x_1, \ldots, x_c\}$ over $b_v$ and the output is $x_{c^*} = f_m(v)$ where $c^* \in \{1, \ldots, c\}$.

The $x_{c^*}$ represents the class in which $b_v$ falls, being based on model $m$.

Two important classification algorithms are Naïve Bayes and hyperplane decision-based classifier.

Naïve Bayes. The model $m$ of this classifier is based on probabilities: the probability that class $x_i$ occurs is $\{p(X = x_i)\}_{i=1}^c$ and the probability that the element $v_j$ of $v$ occurs in the particular class $x_i$. The classification function is:

$$\begin{aligned} x_{c^*} &= \max_{i \in \{1, \ldots, c\}} p(X = x_i | V = v) \\ &= \max_{i \in \{1, \ldots, c\}} p(X = x_i, V_1 = v_1, \ldots, V_n = v_n) \end{aligned} \tag{2}$$

In order to obtain the second equality Bayes rules was applied.

In biometrics, the Naïve Bayes will help us to classify the biometric data and to obtain a faster time for processing. The classification is done based on the biometric characteristics.

Hyperplane decision. The model $m$ contains $c$ vectors in $\mathbb{R}^n$ ($m = \{m_i\}_{i=1}^c$) and the classification function is [8]:

$$x_{c^*} = \max_{i \in \{1, \ldots, c\}} \langle m_i, v \rangle, \tag{3}$$

where $\langle m_i, v \rangle$ represents the inner product between $m_i$ and $v$, in a hypothesis space $H$ having defined an inner product $\langle \cdot, \cdot \rangle$.

In biometrics, the hyperplane decision will help us to identify and to classify in a much better way the biometrics 3D representations of face recognition.

### 2.1.2. Cryptography

Cryptosystem. It has more components: plaintext space $P$, ciphertext space $C$, key space $K$, and encryption functions $E = \{E_k | k \in K\}$, and decryption functions $D = \{D_k | k \in K\}$:

$$\forall e \in K \ \exists d \in K \text{ such that } D_d(E_e(m)) = m, \ \forall m \in P \tag{4}$$

Encryption scheme. It represents a particularization of a cryptosystem. There are two types of encryption schemes: *symmetric key* schemes (in which the same key is used for encryption and also decryption) and *public key* encryption schemes (in which a public key is used to encrypt messages and a private key is used to decrypt messages).

Homomorphic encryption. It is a special type of encryption which allows to apply functions over encrypted message, resulting also an encrypted result, which, when decrypted it is the same as applying the same function over unencrypted message. This is a powerful cryptographic technique, because it increases the security of data, as the operations are applied on encrypted data, resulting an encrypted output, which will be decrypted only the users that own the decryption key. Unfortunately, at this moment a *fully homomorphic encryption* scheme (that allows to apply *any* function on encrypted data) does not exist, because the existing computational capabilities are overwhelm. An example

of partial homomorphic encryption is RSA cryptosystem [9], where the homomorphic operation is multiplication:

$$E(m_1) \cdot E(m_2) = m_1^r m_2^r \bmod n = (m_1 m_2)^r \bmod n = E(m_1 \cdot m_2), \tag{5}$$

where the public key is modulus $n$ and the exponent is $r$, and encryption function is

$$E(m) = m^r \bmod n \tag{6}$$

Applying in biometrics, the function has to be changed in order to allow the operations on bits to be applied for each bit separately.

### 2.2. Techniques

In this section, we present the constructions of the above classification techniques proposed and improved by the authors of [7], such that they could be applied on encrypted data. The authors of [7] have shown that their methods are successfully applied on large real datasets, including in face detection, which became widely used in biometric authentication.

### 2.2.1. Auxiliary Algorithms

In [7] the cryptosystems that have been used are Quadratic Residuosity [10] (where $P = \mathbb{Z}_2$) and Paillier cryptosystem (where $P = \mathbb{Z}_N$ and $N$ is modulus of Paillier) [11]. Further, notation $(b)_{QR}$ means the bit $b$ is encrypted with Quadratic Residuosity, $(m)_{PA}$ means the integer $m$ is encrypted with Paillier, $SK_{QR}$ and $PK_{QR}$ are secret and public key for Quadratic Residuosity and $SK_{PA}$ and $PK_{PA}$ are secret and public key for Paillier.

The entities implied in these sections are two parties A and B for building blocks and C (client) and S (server) for classifiers.

Authors of [7] have defined some auxiliary operations: comparison with unencrypted inputs, comparison with encrypted inputs, reversed comparison over encrypted data, negative integers comparison and sign determination, which will be used in protocols defined below. The Algorithm 1 will demonstrate how the encryption over biometric data can work bit by bit.

---

**Algorithm 1**. Max over encrypted data [7]

---

**Input A**: $k$ integers encrypted using Paillier $((a_1)_{PA}, \ldots, (a_k)_{PA})$, the length $l$ of $a_i$ (in bits), $PK_{QR}$ and $SK_{QR}$
**Input B**: $SK_{PA}$, $PK_{PA}$, the length $l$ in bits
**Output A**: $\max a_i$

A: generate random permutation $\pi$ over $\{1, \ldots, k\}$
A: $(max)_{PA} := a_{\pi(i)}$
B: m:= 1
**for** i = 2 **to** k **do**
$b_i = \max \leq a_{\pi(i)}$
A: randomly generate integers $r_i$, $s_i := \left(0, 2^{\lambda+l}\right) \cap \mathbb{Z}$
A: $(m_i')_{PA} := (max)_{PA} \cdot (r_i)_{PA} \triangleright m_i' = \max + r_i$
A: $(a_i')_{PA} := \left(a'_{\pi(i)}\right)_{PA} \cdot (s_i)_{PA} \triangleright a_i' = a_{\pi(i)} + s_i$
A: send $(m_i')_{PA}$ and $(a_i')_{PA}$ to B
**if** $b_i$ is true **then**
B: $m := i$
B: $(v_i)_{PA} := Refresh(a_i')_{PA} \triangleright v_i = a_i'$
**else**
B: $(v_i)_{PA} := Refresh(m_i')_{PA} \triangleright v_i = m_i'$
**end if**
B: send $(v_i)_{PA}$ to A
B: send $(b_i)_{PA}$
A: $(max)_{PA} := (v)_{PA} \cdot \left(g^{-1} \cdot (b_i)_{PA}\right)^{r_i} \cdot ((b_i)_{PA})^{s_i} \triangleright max = v_i + (b_i - 1) \cdot r_i - b_i \cdot t_i$
**end for**
B: send $m$ to A
A: output $\pi^{-1}(m)$

---

In Algorithm 2 will show how to change an encryption scheme $E_1$ into and encryption scheme $E_2$, both having the same plaintext size $M$. The authors of [7] supposed that $E_1$ and $E_2$ are additively homomorphic, and semantically secure. In the above algorithm $(b)_i$ means $b$ is encrypted using encryption scheme $E_i$, $i \in \{1, 2\}$.

---

**Algorithm 2**. Change encryption scheme [7]

---

**Input A**: $(c)_1$, $PK_1$ and $PK_2$
**Input B**: $SK_1$, $SK_2$
**Output A**: $(c)_2$

A: pick $r \in M$
A: send $(c\prime)_1 := (c)_1 \cdot (r)_1$ to B
B: decrypt $(c')_1$ and re-encrypt with $E_2$
B: send $(c')_2$ to A
A: $(c)_2 = (c\prime)_2 \cdot (r)_2{}^{-1}$
A: output $(c)_2$

---

The authors of [7] proved that these algorithms are secured in 'honest but curious' model.

### 2.2.2. Naïve Bayes

In order to apply Naïve Bayes on encrypted data, there are needed more transformations. In [7], it was working with the logarithm of the probability distribution:

$$x_c^* = \max_{i \in \{1,\ldots,c\}} \log p(X = x_i | V = v)$$

$$= \max_{i \in \{1,\ldots,c\}} \left\{ \log p(X = x_i) + \sum_{j=1}^{n} \log p(V_j = v_j | C = c_i) \right\} \tag{7}$$

The two types of auxiliary table were used:

- One table in which are stored values $P(i) = \lceil K \log p(X = x_i) \rceil$
- One table for every feature j and class $i : T_{i,j}(q) \approx \lceil K \log p(Y_j = q | V = v_i) \rceil$, $\forall q \in D_j$

where $D_j$ represent the domain of possible values of $v_j$, and $K \in \mathbb{N}$ is a constant.

Now to apply Naïve Bayes on encrypted data, the client needs to compute $(p_i)_{PA} = (P_i)_{PA} \prod_{j=1}^{n} \left( T_{i,j}(v_j) \right)_{PA}$, and then uses Algorithm 1 to obtain $\max p_i$.

### 2.2.3. Hyperplane Decision

For hyperplane decision the things are quite simple, because the client will calculate $(\langle m_i, v \rangle)_{PA}$, $i \in \{i, \ldots, c\}$, using Algorithm 3 for inner product, and then Algorithm 1 is used to compute *max* on encrypted inner product.

---

**Algorithm 3**. Private inner product [7]

---

**Input A**: $a = (a_1, \ldots, a_d) \in \mathbb{Z}^d$, $PK_{PA}$
**Input B**: $b = (b_1, \ldots, b_d) \in \mathbb{Z}^d$, $SK_{PA}$
**Output A**: $(a, b)_{PA}$

B: encrypt $b$
B: send $(b_i)_{PA}$ to A
A: compute $(v)_{PA} = \prod_i (b_i)_{PA}^{x_i} \; mod \; N^2 \; \rhd \; v = \sum b_i a_i$

A: re-randomize
A: output $(v)_{PA}$

---

*2.3. Data Clustering in Cloud Computing*

2.3.1. Fixed-Width Clustering Algorithm

Clustering is a machine learning technique (included in unsupervised learning) in which a set of objects is partitioned into groups called clusters. Different from classification, in clustering there are no predefined clusters, thus the algorithm needs to find relationships or similarities between objects [12].

Fixed-width clustering (FWC) algorithm is based on a distance measure. The steps of FWC are the following:

1.  From a given dataset $D$ with an established cluster width $w$, generate a random set of $m$ clusters: $C_i$, $i \in \{1, \ldots, m\}$.
2.  Compute Euclidean distance between every point $p_j$, $j = \{1, \ldots, n\}$ and every cluster $C_i$, using the formula:

$$d_{ij}(c_i, p_j) = \sqrt{(c_{ix} - p_{jx})^2 + (c_{iy} - p_{jy})^2} \qquad (8)$$

3.  If $d_{ij}(c_i, p_j) \leq w$, then $p_j$ belongs to $C_i$ cluster; adjust the centroid of $C_i$ by computing the mean of the points that $C_i$ contains at this moment, using the formula ($n$ is the number of points in $C_i$):

$$centroid(C_i) = \left( \frac{p_{1x} + \cdots + p_{nx}}{n}, \frac{p_{1y} + \cdots + p_{ny}}{n} \right) \qquad (9)$$

4.  If $d_{ij}(c_i, p_j) > w$, then $p_j$ is the new centroid of $C_i$.
5.  Reiterate steps 2, 3, 4 until the end of $D$.

2.3.2. MapReduce

MapReduce [13] is a programming model for large datasets processing, which works exclusively on (*key, value*) pairs. It consists in three steps: *map, shuffle, reduce*, and the user needs to define *map* and *reduce* functions. The basic idea is that *map* function takes as input a set of (*key, value*) pairs and outputs an intermediary set of (*key, value*). These outputs are processed by *shuffle* function, which groups all intermediary values corresponding to an intermediary key, and sends them to *reduce* function. The *reduce* function will try to join these values in order to decrease the number of values. Usually just one value will result for the input key per *reduce* invocation.

A well-known software framework is Hadoop MapReduce [12] that allows processing in parallel large datasets (multi-terabytes of data). We will not give more details, for a comprehensive view please read [12].

2.3.3. FWC Algorithm with MapReduce

In [14] the authors propose a distributed version of FWC algorithm, using MapReduce on a large number of virtual machines (VMs), as follows:

-   Inputs: dataset D and the set of clusters $C_1, \ldots, C_m$
-   Partitioning: the $N$ points of $D$ are allocated to the $M$ available VMs (if $\frac{M}{N}$ is not integer, then the remaining points are allocated to the last VM).
-   Map function. The input is dataset $D$ encrypted and kept into Hadoop Distributed File System (HDFS) as (*key, value*) pairs, where *key* represents the position of *value* in a data file and *value* represents the encryption of numerical of the data point. The data files are global and sent to all mappers. The *map* function in proposed model computes the squared Euclidean distance (in order to shun the square root):

$$E(d_{ij}(c_i, p_j)) = (E(c_{ix}) - E(p_{jx}))^2 + (E(c_{iy}) - E(p_{jy}))^2 \qquad (10)$$

The *output* of map is a set of (*key*, *value*) pairs, where *key* is the position of *value* into a data file and *value* is the distance $E(d_{ij}) \overset{\text{def}}{=} E(d_{ij}(c_i, p_j))$. Note that $E(v)$ means value $v$ is encrypted using function $E$.

- *Reduce* function. The output of a *map* function becomes the input for a *reduce* function. The *reduce* function needs to find a minimum distance between every point $E(p_j)$ and every centroid $E(c_i)$ and then to put data point $E(p_j)$ into corresponding cluster (the one that corresponds to the minimum distance) [12].

Next, we present the pseudo-code algorithms for *map* and *reduce* functions as the authors provide them in [14].

---

**Algorithm 4**. Map function for distributed version for FWC [14]

---

**Input:** encrypted dataset $E(D)$
**Output:** ⟨*key*, *ctxt(value)*⟩ → ⟨*index*, *encrypted distance* $E\left(d_{ij}\right)$⟩

Initialization: Choose a random set of clusters $c_1, \ldots c_m$ from a given dataset $E(D)$
index = 0
for (*i*=0 to *D.length*) do
for (*j*=0 to *c.length*) do

$$E\left(d_{ij}\right) = computeDist\left(E\left(p_j\right), E(c_i)\right)$$
$$index = i + j$$

end
end
End
Take index as key
Construct value as an encrypted numerical value $E\left(d_{ij}\right)$
Output ⟨*key*, *ctxt(value)*⟩ pair

---

**Algorithm 5**. Reduce function for distributed for FWC [14]

---

**Input:** ⟨*index*, *encrypted distance* $E\left(d_{ij}\right)$⟩
**Output:** ⟨*ctxt(key)*, *ctxt(value)*⟩ → ⟨$E(c_i)$, $E\left(p_j\right)$⟩

Initialization: $E(minDis)$
for (*i*=0 to *D.length*) do

$$E(minDis) = \min\left(d_{i1}, \ldots, d_{ij}\right)$$

if $E(minDis) \leq w$ then

$$assign\left(E\left(p_j\right), E(c_i)\right)$$
$$update(E(c_i))$$

else

$$createNewCluster\left(E\left(p_j\right)\right)$$

end
end
Take $E\left(p_j\right)$ as key
Construct value as a numerical value $E(c_i)$
Output ⟨*ctxt(key)*, *ctxt(value)*⟩ pair

---

We can easily see the potential of this approach of FWC algorithm in biometry. For example, it could be used on large datasets of human face datasets in order to find different groups among the subjects by analyzing the face images. An advantage of the proposed model in [14] is that computations are made on encrypted data, assuring data privacy.

## 3. Biometrics and Authentication Mechanisms: How They Are Working?

For our framework is very important to understand the main mechanisms that a biometric authentication process is used in real life. For this we have to examine all three phases: enrollment, identification and verification [15–21].

Before implementing any solution in a practical system, we need to understanding that in a practical system, a biometric system is facing a number of other issues that should be taken into consideration before proceeding with implementation [22–36]. These issues are:

- Performance—consists of recognition accuracy and speed. The resources that are allocated to achieve the desired accuracy and speed [37].
- Acceptability—represents the threshold at which the people are willing to accept the use of a specific biometric characteristic in their daily lives [37,38].
- Circumvention—this consist in how easily the system can be tricked with fraudulent methods [39,40].

### 3.1. What Is Happening in Enrollment Mode?

The goal of enrollment mode (Figure 1) is to register the user with its own biometric characteristics. The biometric characteristics are stored in database as templates. Is very important to store these templates encrypted using the framework proposed and discussed later.

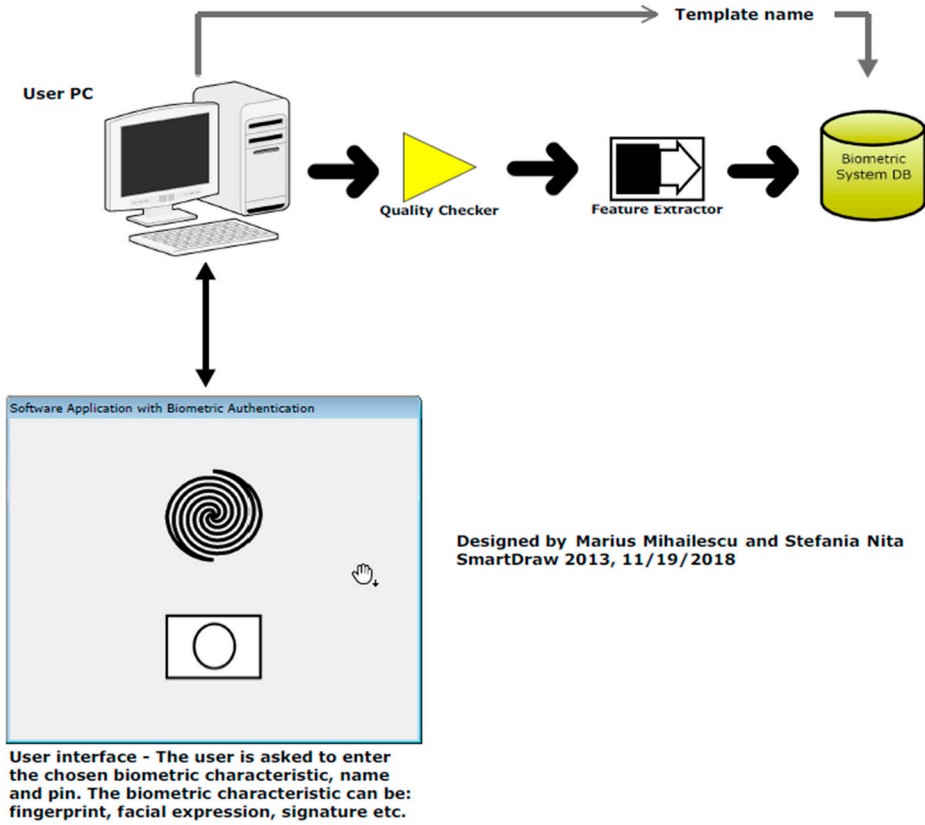

**Figure 1.** Enrollment phase (mode).

The mode has two important steps: Quality Checker and Feature Extractor. The purpose of these two steps are critical and extra security measures need to be taken in order to have a secured environment for the user during the scanning process.

The purpose of the Quality Checker is to make sure that the quality of the biometric characteristic is properly done and the checking of the quality is done properly using the proper template in order to be interpreted by the Feature Extractor module.

*3.2. What Is Happening in Identification Mode?*

In identification mode (Figure 2) the system has a clear goal to carry out the one-to-many comparison meant to set up the individual identity. With other words, the user's identity and the templates that are stored in the database are compared and based on the result a decision will be made. The purpose of the identification process is to answer to the question "Who am I?" If we are talking about the implementation process, we can say that they are time consuming for deploying and needs a huge amount of time for processing in order to find the proper match within the database [1].

The general goal is for the system to be able to recognize the person by searching through the templates of all the users that are stored in the database and to find a match [41].

Identification is a very critical component when we are talking about negative recognition application in which the system will establish whether the person is who she denies to be. There is a very important purpose of negative recognition which consist in preventing a single person from using multiple identities [36,42–54].

Identification can be used in positive recognition for convenience. Traditional methods of personal recognition such as PIN, passwords, keys and tokens may work for positive recognition, negative recognition can only be settled through biometrics.

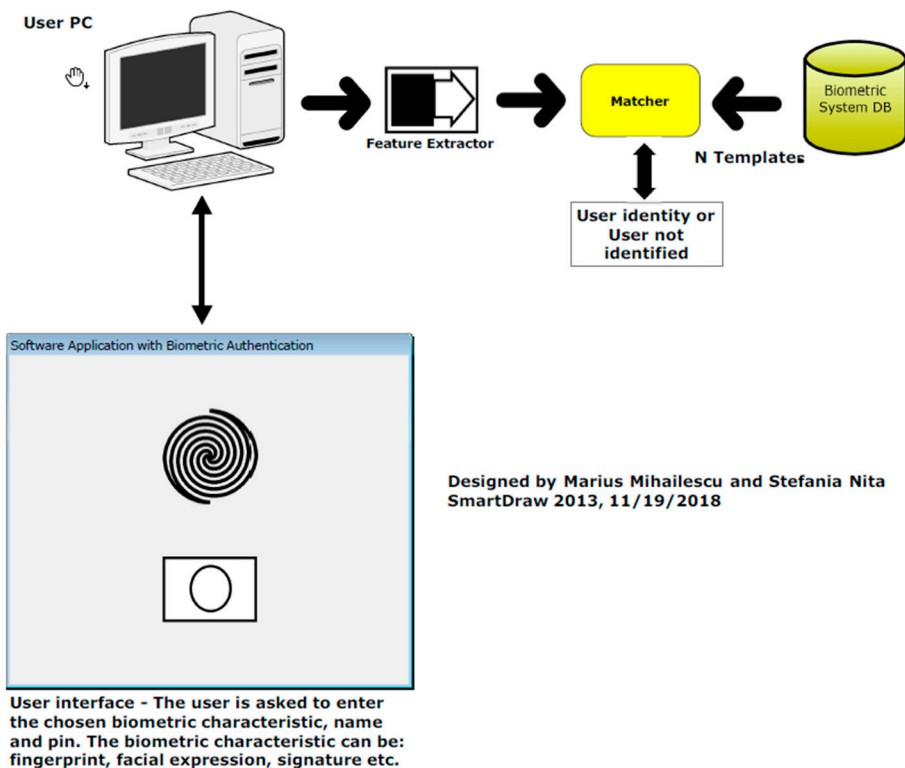

**Figure 2.** Identification phase (mode).

*3.3. What Is Happening in Verification Mode?*

In verification mode (Figure 3), the system is carrying out the one-to-one comparison. This type o comparison is used to set up the individual identity. The user is claiming that identity and the system

has the duty to verify if the claim is genuine or forged. The goal of the verification process is to answer to the question "Am I who I say I am?" [1].

The mechanism consists in validation of the person identity by comparing the captured biometric data with the template(s) stored in the database. In this phase is very important how we store the templates in the database and how they are compared. Is very important to store the biometric templates encrypted and the comparing process to be done encrypted by using a complex encryption mechanism as they are described below.

The identity verification is used to get positive recognition where the goal is to prevent different people to use the same identity.

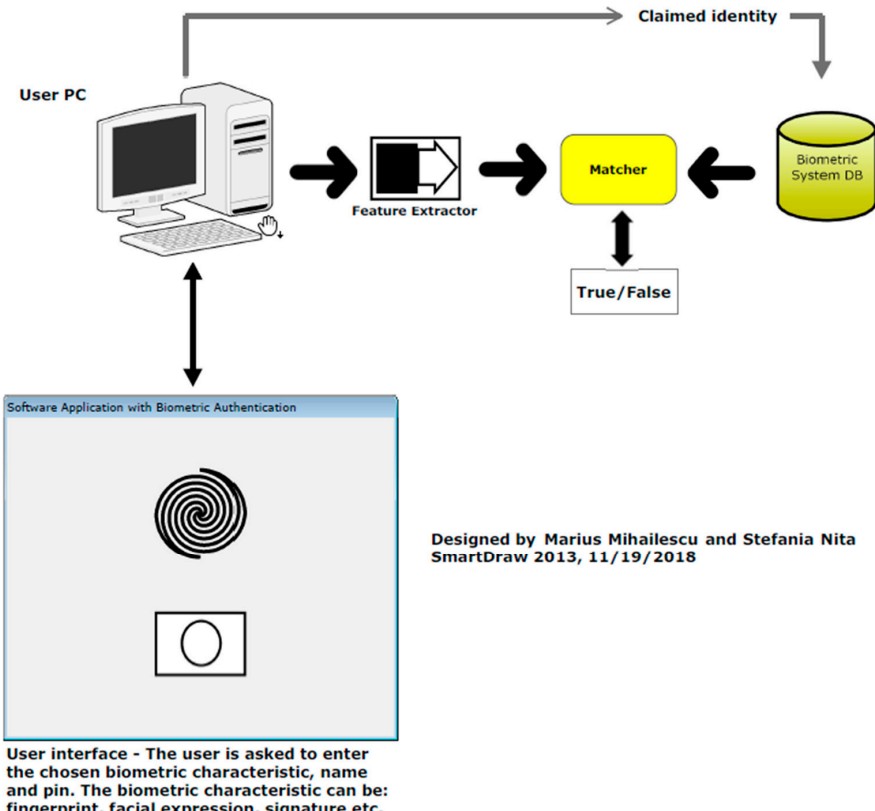

**Figure 3.** Verification phase (mode).

An interesting question raised by different professionals is "What measurements for the biological characteristics is making them to be qualified as a biometric?" Most of the human physiological and/or behavioral characteristics used in the system for authentication as long as they are satisfying the following requirements:

-　　Performance: this requirement is quite important as the characteristic should be enough invariant. The respect has to be assured for the matching criterion over a period.
-　　Distinctiveness: by choosing two persons should be sufficient different in terms of the characteristic.
-　　Universality: the criteria consist in its unique characteristic that has to be for each person.
-　　Collectability: the requirement is a metric that is quantitatively measured.

When we are dealing with a real life biometric system, there is a number of issues that has to be taken into consideration in order to take a complete advantage of the full system [39,40] and to combat those security issues that could occur on different section:

- Performance, the accuracy and speed, two main characteristics that refers to the achievable recognition, are required to achieve the desired recognition accuracy and speed. Also, operational and environmental factors are affecting the accuracy and speed;
- Acceptability, a factor that will indicates which people are willing to accept the use of a particular biometric identity in terms of characteristic using in a daily life;
- Circumvention reflects how easy is to fool a system using different methods meant to steal data and to corrupt the integrity of the data.

As we will be able to see, the following content of the present work paper will cover the most important aspects of security flaws that are raised by each of the components of the biometric system. The work paper will show and demonstrate theoretical and practical two main benefits: (1) we will demonstrate how we can protect the integrity of the biometric data using Machine Learning Classification and what benefits we can obtain. Another benefit, and (2) applying chaos-based cryptography over encrypted biometric encrypted data. The two main methods for securing and guaranteeing the integrity of the biometric encrypted data will be demonstrated in a professional system architecture based on Hadoop and Data clustering.

## 4. Results

This section will describe the proposed solution, which has been practically implemented already, and the results obtained where positive in encrypting data. The idea was developed and presented in details in [55,56].

The below scheme can be adapted with success for all the mentioned algorithms mentioned above. All the algorithms were been implemented with success in C#, Java, C++, Python, and Haskell. The source code will be available at the link mentioned below.

The proposed framework (Figure 4) is designed on three layers: extension layer, application layer, and framework layer. Each layer is designed in such way to be adapted easily to the application that is developed taken into consideration the requirements of the platform on which the application will be integrated and deployed.

*Description of layers*

In *extension layer* we have the all the necessary modules that contains the implementations of the algorithms mentioned in the previous section. Each module from extension layer corresponds to each module from framework player.

*Modules* contains all the references and interfaces that the *libraries* module (from framework layer) serves as an access for all the programming concepts (such as classes and interfaces, methods and functions). Below you can see the example for each of the programming language.

Java

```
import BioCrypto.ModuleName.*;
```

C#

```
using BioCrypto.ModuleName;
```

C++

```
#include "biocrypto.h"
```

Python

```
import biocrypto as bc
```

Haskell

```
import biocrypto
```

Cryptographic Algorithms module contains the implementations of the following cryptographic algorithms: AES (Advanced Encryption System), SHA-256, RC4, RC5, RC6, MARS, BLOWFISH,

TWOFISH, THREEFISH, RSA (Rivest-Shamir-Adleman), Elliptic Curve, and Diffie Hellman. The access for the implementation is done through *Interfaces* module from framework layer. In this way the functions can be adapted easily to platform requirements. The cryptographic algorithms implementation can be found.

Machine Learning algorithms module contains the implementations of all the algorithms that are necessary for evaluation of the encrypted data. This module can be seen as optional and it can be delivered during the implementation as a plugin.

Data Clustering algorithms module contains all the algorithms necessary to adapt the security requirements to cloud computing and Hadoop environment. In this module we will find the implementations of three algorithms: Fixed-width clustering algorithm, Map Reduce methods and functions to identify the security requirements, and FWC Algorithm based on Map Reduce. When is used in the developing and implementation process, after invoking the proper module, the Other Interfaces module from framework layer, the interfaces need to be customized accordingly.

Apache Hadoop and Map Reduce module contains the necessary functions and implementations for adapting the biometric devices and hardware devices if it is required. The access is done through *Biometric Modules* module which contains the interfaces and functions necessary to be invoked during the developing phase.

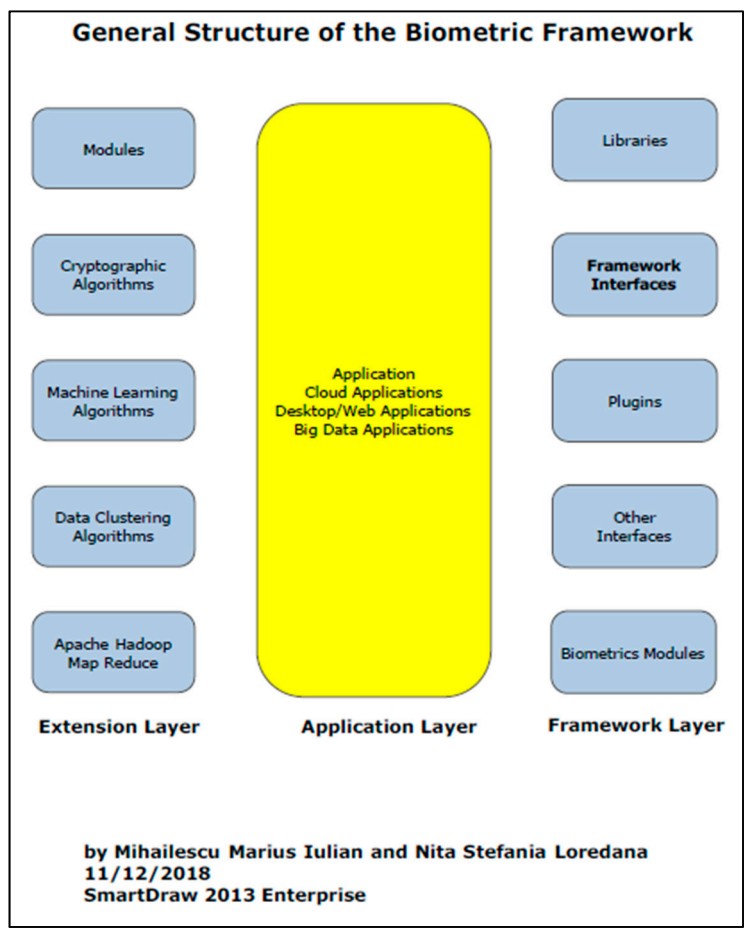

**Figure 4.** The General Structure of the Biometric Framework.

In Figure 5 we show how to use the framework in the process of implementation. *Biometric Utilities* module is based on four modules: Components, Services Software Stack, Drivers, and Boards. All the modules are related and interconnected within the framework. Below, we have a short description of the four modules.

Components module represents all the drivers and libraries necessary for the biometric devices and biometric hardware equipment. *Services Software Stack* module represents the collection of all the services that can be invoked through different stages of the software or web applications development. *Drivers* module contains all the drivers and setup kits for different biometrics hardware devices based on boards, such as signature pads or iris recognition device. *Boards* module contains functions and abstractions of the functions for Hadoop environment.

The below example shows in Figure 6 how the biometric templates should be stored in the database. The scheme is just an example of how to use the framework and how to invoke the functions.

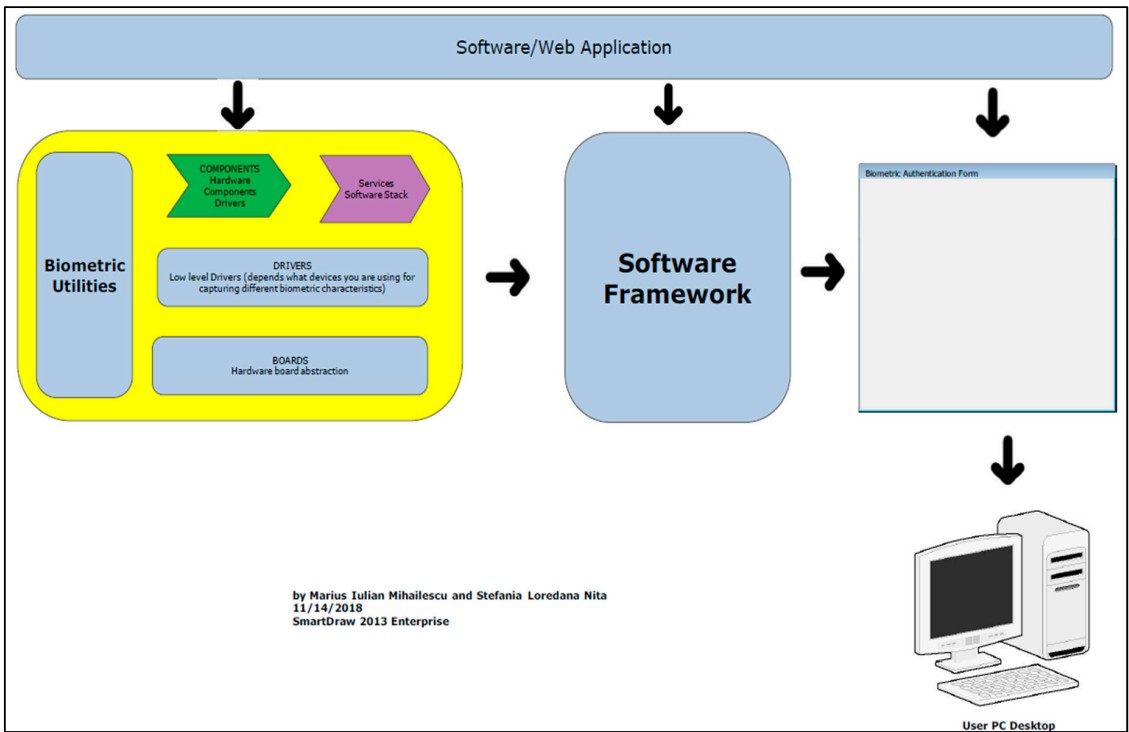

**Figure 5.** General workflow on how to use the framework.

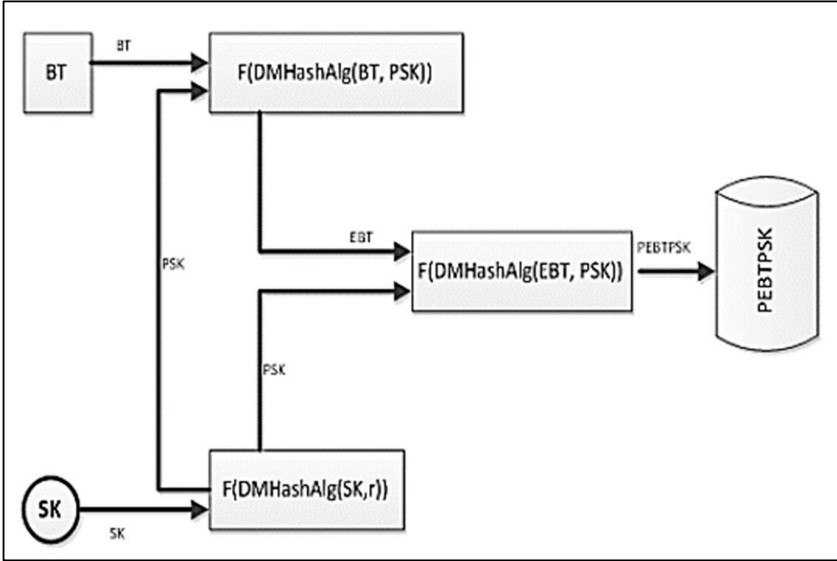

**Figure 6.** Data integrity checking and validation for biometric data.

The scheme has been created in a very flexible manner giving the possibility to be adapted accordingly to any type of algorithms especially for those ones from machine learning field.

The algorithm has two components:

1.  The session key algorithm, and
2.  The scheme used for enrollment with data integrity checking and validating for the biometric data.

In Figure 8 we can see that the permutation functions are applied on all the components that play a role in the enrollment scheme. The permutation function will allow the original value to be re-arranged in such a way that will be very difficult to understand something from permuted value. The original values influence the encrypted system. The content of data could be protected using the complex active action, which comes from the characteristic of the system mentioned above. The behavior will result into a random series, which could be utilized to data encryption from secret communication. Thus, an appropriate key controls encryption or decryption of data content. Machine learning is also an appropriate choice for using a hash function due to one-way property. The model described in Figure 6 stores the encrypted Biometric Template using Session Key. A session key is the process that generates a random encoding and decoding key which ensures the privacy of a session of communications. A similar example can be found in [10,11].

As some notations for the scheme presented in Figure 4, the followings have to be considered:

*   BT—biometric data.
*   SK—represents the session key, which is generated using a one of the algorithms, which were presented in Chapter 2 and 3 and combined, with elements of machine learning described in Section 2.1.1. The BT contains the biometric vector $(b_v)$ as we have discussed previous.
*   PSK—represents the permuted session key, which is used to generate the extended version of permuted transformation of the session, key (SK). F(SK) represents a function used for permutation which can be used with any hash function generated based on the main ideas presented above.
*   EBT—represents the biometric data, which are encrypted. In order to generate the biometric template encrypted, the hash function construction is applied F(DMHashAlg(BT, PSK)). The hash function is based on a simple XOR function and both functions F(DMHashAlg(SK,r)) and F(DMHashAlg(EBt, PSL)) functions are used together beside the hash functions with the permutation of the bits SK, EBT, and ESK.
*   PEBTPSK—the permuted biometric data and also the permuted session key (SK) will be used to generate the final step in order to concatenate the biometric template F(DMHashAlg(EBT, PSL)). In order to assure the decryption process, the biometric pattern is using the functions and session key, which has been used to encrypt the template.

In the end, the presented idea represents a new scheme for assuring the data integrity of the biometric data. The algorithm has been implemented with success and it was tested with positive results on a set of 700 unique biometric data of 523 subjects.

To access the source code of the application, please, visit the following web address: https://www.researchgate.net/project/Biometrics-Analysis-Tool.

## 5. Comparison with Other Proposed Methods and Discussion

Comparing with many existing cryptographic algorithms, we have selected several algorithms, such as AES (Advanced Encryption System), SHA-256, RC4, RC5, RC6, MARS, BLOWFISH, TWOFISH, THREEFISH, RSA (Rivest-Shamir-Adleman), Elliptic Curve, and Diffie Hellman. These algorithms were selected and compared based on their basis of structure, security, flexibility to expand their limitation in the near future [57,58]. Is very important to understand the security vulnerabilities of the algorithms and based on the security analysis to take the proper decision and to improve the security parameters used by users during their authentication process.

The comparison done in Table 1 is a comparison taken into consideration the measurement of quality. Quality of the encryption process is very important when the biometric templates are encrypted and stored in the database.

**Table 1.** Quality measures.

| Algorithm | Structure | Flexibility and Modification | Known Attacks |
|---|---|---|---|
| DES [59–61] | Balanced Feistel Network | No | Brute Force Attack |
| 3DES [62–66] | Feistel | Yes, Extended from 56 to 168 bits | Brute Force Attack, Chosen Plaintext, Known Plaintext |
| CAT-128 [67] | Feistel | Yes, 128 and 256 bits | Chosen Plaintext Attack |
| BLOWFISH [68] | Feistel | Yes, 64–448 key length in multiplies of 32 | Dictionary attacks |
| IDEA [2,3] | Substitution-Permutation | No | Differential Timing Attack, Key-Schedule Attack |
| AES [60–63,69] | Substitution-Permutation | Yes, 256 key length in multiples of 64 | Side Channel Attack |
| RC4 [64–66,70,71] | Feistel | Yes, 40–2048 bits | Fluhrer, Mantin and Shamir Attack, Klein's Attack, Royal Holloway Attack, NOMORE Attack, Bar-mitzvah Attack |
| RC5 [57] | Feistel | Yes, 0 to 2040 (128 recommended) | Differential Attack |
| RC6 [58] | Feistel | Yes, 128-2048 key length in multiplies of 32 | Bruce Force Attack, Analytical Attack |
| MARS [72] | Type-3 Feistel | Yes, 128, 192, or 256 bits | Meet-in-the-middle Attack |
| TWOFISH [73] | Feistel | Yes, 128, 192 or 256 bits | Differential Attack |
| THREEFISH [2,3] | Feistel | Yes, 256, 512 or 1024 bits (key size is equal to block size) | Rebound Attack, Boomerang Attack |
| RSA [5] | Factorization | Yes, Multi Prime RSA, Multi power RSA | Factoring the Public Key |

Based on the above analysis, compared with our solution described in Section 5 and Figure 8, the scheme proposed for encryption of biometrics templates is much faster and reliable. As a case study, the below table shows a comparison between the scheme proposed in the article and other schemes proposed by other authors.

Below in Table 2 and Figure 7, we shown that Blowfish and AES are having the best performance compared with others. Both of them are known to have better encryption (i.e., stronger against data attacks) than the other two.

**Table 2.** Comparison results with our scheme.

| Algorithm | Megabytes($2^{20}$ bytes) Processed | Time Taken | MB/Second |
|---|---|---|---|
| AES [1] | 256 | 3.976 | 64.386 |
| SHA-256 [55] | 256 | 4.196 | 61.010 |
| RC4 [68] | 256 | 4.817 | 53.145 |
| RC5 [4] | 256 | 5.308 | 48.229 |
| RC6 [62] | 256 | 4.436 | 57.710 |
| MARS [66] | 256 | 4.837 | 52.925 |
| BLOWFISH [70] | 256 | 5.378 | 47.601 |
| TWOFISH [17] | 256 | 4.617 | 55.447 |
| THREEFISH [61] | 128 | 5.998 | 21.340 |
| RSA [62] | 128 | 6.159 | 20.783 |
| ELLIPTICCURVE [71] | 64 | 6.499 | 9.848 |
| DIFFIE HELLMAN [57] | 64 | 6.389 | 8.763 |

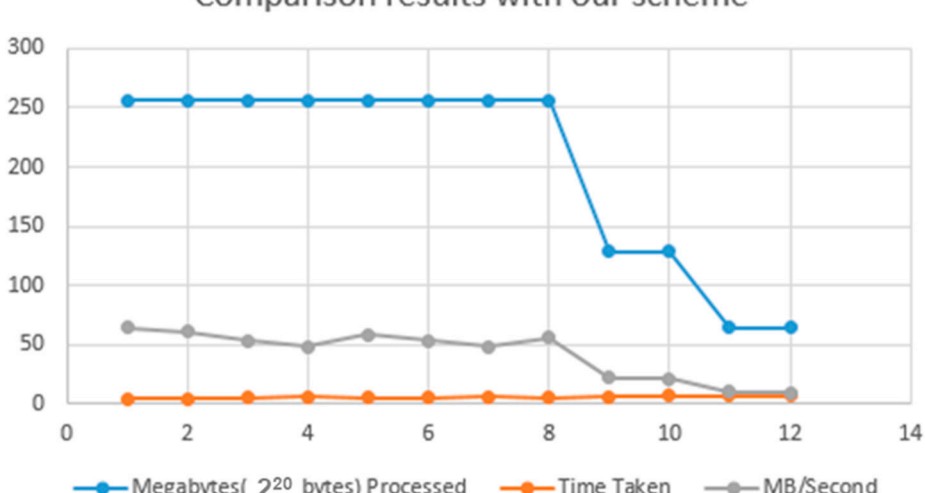

**Figure 7.** Diagram with comparison results with the scheme proposed.

Table 3 is showing the results of the experiments that we have conduct. In Figures 5 and 8 we have depict the execution times with the configuration on which we have done the tests.

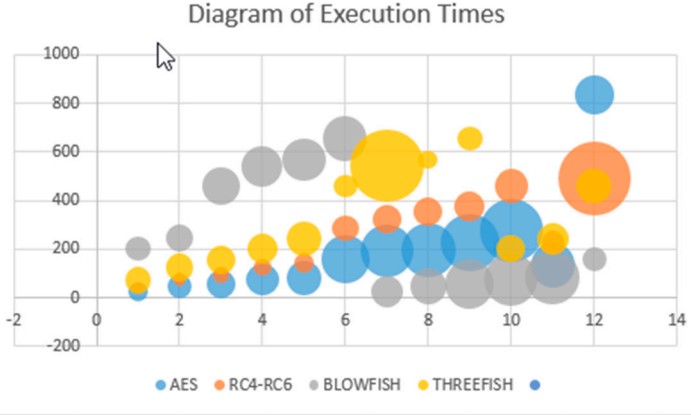

**Figure 8.** Execution Times

**Table 3.** Execution times (in seconds) comparison with our scheme on Intel Core i7-4510U CPU @2.00 GHz 2.60 GHz with 8 GB RAM.

| Input Size (bytes) | AES | SHA-256 | RC4-RC6 | MARS | BLOWFISH | TWOFISH | THREEFISH | RSA |
|---|---|---|---|---|---|---|---|---|
| 20,527 | 24 | 72 | 39 | 19 | 202 | 125 | 72 | 136 |
| 36,002 | 48 | 123 | 74 | 35 | 243 | 143 | 123 | 158 |
| 45,911 | 57 | 158 | 94 | 46 | 461 | 285 | 158 | 162 |
| 59,852 | 74 | 202 | 125 | 58 | 543 | 324 | 202 | 176 |
| 69,545 | 83 | 243 | 143 | 67 | 569 | 355 | 243 | 219 |
| 137,325 | 160 | 461 | 285 | 136 | 655 | 378 | 461 | 108 |
| 158,959 | 190 | 543 | 324 | 158 | 24 | 202 | 543 | 1036 |
| 166,364 | 198 | 569 | 355 | 162 | 48 | 243 | 569 | 72 |
| 191,383 | 227 | 655 | 378 | 176 | 57 | 461 | 655 | 123 |
| 232,398 | 276 | 799 | 460 | 219 | 74 | 543 | 202 | 158 |
| Average Time | 134 | 383 | 228 | 108 | 83 | 569 | 243 | 202 |
| Bytes/s | 835 | 292 | 491 | 1036 | 160 | 108 | 461 | 243 |

## 6. Conclusions

Applying cryptographic mechanisms and machine learning over biometric data is not an easy task to accomplish. This fact can be due to the high complexity of how the biometric data are scanned and read from the user and transfer into the system. Every time that the evolution of technology

is making important advances the complexity of assuring the security and integrity of the data is becoming a real pain for developers and designers of authentication systems on biometrics.

We have proven that applying machine learning techniques and cryptography mechanisms can be a task that can be accomplished. The most important aspect on which we have focused in this work paper is how the parameters can be represented and adapted within the algorithms and techniques used in cryptography and machine-learning. The complexity of the algorithms and the time processing represent a problem but not so critical at this time for any of the system configuration based on the highest requirements possible.

The algorithms presented in Sections 2–4 where implemented with success and we have obtained positive results.

The advantages of the current scheme proposed and methodology for developing applications for Hadoop using biometrics as authentication are:

Fast and reliable scheme for biometric templates encryption;

- Fast adaptable framework for different platforms;
- Reliable implementation of cryptographic algorithms and machine learning.

All the results can be viewed at https://www.researchgate.net/project/Biometrics-Analysis-Tool. The software used in analysis and simulation is presented at the mentioned web site. Due to copyrights, we did not make the source code available at this moment.

The new challenges from the last two years represents a very important alarm signal for both, academy and business environments. The security threats and gaps in assuring the privacy and integrity found, represents one of the most important occasion from which we have to take the maximum advantage in creating new theoretical and practical security frameworks.

Below we have underlined the main new challenges that in our opinion will create new research directions for security and cryptography field, such as:

- Using Machine Learning Classification over the encrypted biometric data;
- Encryption of biometric data in a Data Clustering environment;
- Encryption of biometric data using Chaos-based cryptography;

In order to be able to follow the ideas and to understand how they will be applied in a real environment, we need to understand the main four modules of the biometric system and their vulnerabilities.

The mentioned challenges raised above will be treated and the issues will be solved using machine-learning classification applied in cluster environment using authentication-based biometrics.

**Author Contributions:** Formal analysis, V.C.P.; Methodology, V.C.P.; Software, S.L.N. and M.I.M.; Supervision, V.C.P.; Writing—original draft, S.L.N.; Writing—review & editing, M.I.M.

**Funding:** This research received no external funding.

**Conflicts of Interest:** The authors declare no conflicts of interest.

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
