# Peer review of "Security and Cryptographic Challenges for Authentication Based on Biometrics Data"

_cryptography, doi:10.3390/cryptography2040039_

Reviewer 1 Report

  1. The references are not enough. The authors are suggested to cite more papers in recent 3-5 years.

  2. The abstract should indicate the motivations, method and your contributions. The abstract should be reorganized.

  3. The related works do not give a clear framework. Why you do this research? What are the deficiency of the past research?

  4. The analysis is too rough. You should discuss the performance, computation cost and attacks etc. issues.

  5. Make a comparison with related works.

  6. The current conclusion presentation does not show what are your contributions and solve some hard problems.

Author Response

Review/Observation

Authors notes

Checked (Yes/No)?

1.The   references are not enough. The authors are suggested to cite more papers in   recent 3-5 years.

References list has been improved to 73 references   items.

Yes

2.The   abstract should indicate the motivations, method and your contributions. The   abstract should be reorganized.

Improved as requested by pointing out exactly what has been   requested.

Yes

3.The   related works do not give a clear framework. Why you do this research? What   are the deficiency of the past research?

The presentation of the framework has been   improved accordingly. See Section 4 – Results from Page 12.

Yes

4.The   analysis is too rough. You should discuss the performance, computation cost   and attacks etc. issues.

Improved as requested by providing performance and computation, focusing   also on the attacks. See Section 5 – Comparison with other proposed methods.   Discussion Page 15.

Yes

5.Make   a comparison with related works.

Done accordingly. See Page 16. The comparison has   been done on quality measures.

Yes

6.The   current conclusion presentation does not show what are your contributions and   solve some hard problems.

Improved and explained properly all the points indicated in the   review

Yes

Reviewer 2 Report

The article is on a decent level but the topic is interesting. A lot of aspects should be improved. So, I can recommend to accept paper after major revision.

In order to improve manuscript, I suggest a few recommendations:

R1: Please write clearly in the abstract about: 1) work objective, 2) methods, 3) results (present and summarize the obtained result), and 4) conclusion.

R2: In section introduction, please write clearly: 1) what is the goal of the study, 2) write what is the contribution of the authors, and 3) identify motivations for undertaking research.

R3: In "Related Work" section it is worth mentioning interesting research about biometrics based on touch screen gestures (eg article: "Person recognition based on touch screen gestures using computational intelligence methods").

R4: Please significantly expand the review of the literature concerning biometrics for at least 30 references.

R5: Please significantly expand the "4 Our proposed solution" section. In which should be accurately describe the results obtained and describe the materials and methods used.

R6: Please add the "Discussion" section in which the conclusions should be described based on the results obtained. And also, in this section, should be a comparison of the proposed method to the methods of other authors.

R7: Please consider replacing section names to: 1) introduction, 2) materials and methods, 3) results, 4) discussion and 5) conclusion.

R8: Please in the conclusion section: 1) list the advantages and disadvantages of the proposed solution, 2) indicate the limitations of work, and 3) indicate directions for further research.

Author Response

Review/Observation

Authors notes

Checked (Yes/No)?

R1:   Please write clearly in the abstract about: 1) work objective, 2) methods, 3)   results (present and summarize the obtained result), and 4) conclusion.

The abstract has been improved and the work   objective and accomplished results within the article have been explained   accordingly.

Yes

R2:   In section introduction, please write clearly: 1) what is the goal of the   study, 2) write what is the contribution of the authors, and 3) identify   motivations for undertaking research.

The introduction has been improved and all the mentioned points were   checked and marked with italics in   Introduction Section. Pages 1-3.

Yes

R3:   In "Related Work" section it is worth mentioning interesting   research about biometrics based on touch screen gestures (eg article:   "Person recognition based on touch screen gestures using computational   intelligence methods").

The related work has been presented in Section 2   in details by presenting all the methods and algorithms proposed by other   authors. Another section, Section 3 – Biometrics and Authentication   Mechanisms. How they are working?, has been added explaining the main phases   of a biometric authentication process which is very important in the   development process of a software/web application based on authentication using   biometric characteristics.

Yes

R4:   Please significantly expand the review of the literature concerning   biometrics for at least 30 references

Improved accordingly and provided more accurate references.

Yes

R5:   Please significantly expand the "4 Our proposed solution" section.   In which should be accurately describe the results obtained and describe the   materials and methods used

The section 4 now is Section 5 – The prosed   solution and methodology has been accurately described and more details have   been provided.

Yes

R6:   Please add the "Discussion" section in which the conclusions should   be described based on the results obtained. And also, in this section, should   be a comparison of the proposed method to the methods of other authors

The Discussion section has been done already within Section 5 being   related. In Page 16 and 17 other methods proposed by other authors have been   mentioned and compared with ours.

Yes

R7:   Please consider replacing section names to: 1) introduction, 2) materials and   methods, 3) results, 4) discussion and 5) conclusion

The names of the sections have been taken into   consideration and they have been renamed as follows:

-         1) introduction with 1. Introduction (page 1);

-         2) materials and methods with Section 2 - Algorithms and Methods Used   (page 2) and Section 3 – Biometrics and Authentication Mechanisms. How they   are working? (page 10);

-         3) results with Section 4- Results (page 12);

-         4) discussion with Section 5 – Comparison with other proposed   methods. Discussion (page 15);

-         5) conclusion with Section 6 - Conclusions

Yes

R8:   Please in the conclusion section: 1) list the advantages and disadvantages of   the proposed solution, 2) indicate the limitations of work, and 3) indicate   directions for further research

Mentioned as requested.

Yes

Round  2

Reviewer 1 Report

The authors have fixed some previous concern issues.

Reviewer 2 Report

The article is ready for publication because the authors have improved the article in accordance with all reviewers comments.